# Ecosystem Organic Carbon Stock Estimations in the Sile River, North Eastern Italy

**Alessandro Buosi** \* , **Yari Tomio** , **Abdul-Salam Juhmani**  **and Adriano Sfriso** 

Department of Environmental Sciences, Informatics and Statistics (DAIS), Ca' Foscari University of Venice, Via Torino 155, 30170 Venice, Italy; yari.tomio@unive.it (Y.T.); abdulsalam.juhmani@unive.it (A.-S.J.); sfrisoad@unive.it (A.S.)

\* Correspondence: alessandro.buosi@unive.it; Tel.: +39-041-234-8621

**Abstract:** River ecosystems are one of the dynamic components of the terrestrial carbon cycle that provide a crucial function in ecosystem processes and high value to ecosystem services. A large amount of carbon is transported from terrestrial to the ocean through river flows. In order to evaluate the contribution of Sile River ecosystem to the global carbon stock, the river ecosystem Organic Carbon (OC) stock was quantified for sediments and dominant submerged aquatic macrophytes (SAMs) during the two sampling periods at three different stations along the Sile River (North Eastern Italy). The total mean ecosystem OC stock was $95.2 \pm 13.8$ Mg C ha$^{-1}$ while those of SAMs ranged from 7.0 to 10.9 Mg C ha$^{-1}$ which accounted for approx. 10% of the total OC stock. The total above-ground biomass retains approx. 90% of the SAM carbon stock, with a mean of $8.9 \pm 1.6$ Mg C ha$^{-1}$. The mean sediment OC stock was $86.6 \pm 14.5$ Mg C ha$^{-1}$ with low seasonal variations among the sites. Indeed, various environmental parameters and hydrodynamics appear to affect the accumulation of OC within the river ecosystem. The results highlight the role that freshwater river ecosystems play in the global carbon cycle, which consequently provide a baseline for future river ecosystem monitoring programs. Furthermore, future studies with additional sites and seasonal surveys of the river will enhance our understanding of the effects of global climate change on the river ecosystem and improve the ecosystem services.

**Keywords:** organic carbon stock; submerged aquatic macrophytes; river ecosystem; global climate change

## 1. Introduction

Rivers represent the most dynamic components of the terrestrial carbon cycle and provide important functions in ecosystem processes. Significant amounts of organic and inorganic carbon typically deliver rivers from the surrounding landscape or originate from the photosynthesis of algae and plants in the water [1]. Despite their dynamic role in the terrestrial carbon stock cycle, river systems and the potential mechanistic controls of OC storage are among the least investigated [2], in comparison to other ecosystems. To date, the limited conducted research suggests that river networks might store a significant proportion of terrestrial carbon [3,4]. Research in ecosystem processing underlines the importance of Organic Carbon (OC) in river systems. Indeed, the fluxes in OC in terrestrial freshwater systems indicate that a marked portion of terrestrial carbon is stored within river networks [2]. River OC fluxes exert an important control of freshwater ecosystems and their ecosystem services. Therefore, understanding physical and biological processes in rivers is crucial for determining the potential impacts of global climate change, land-use, and land-cover changes on OC dynamics within river systems [5].

Different studies highlight the difficulty in identifying the major sources of carbon to freshwater ecosystems, which is critical to properly link the terrestrial and aquatic carbon cycles [6].

Globally, rivers receive about 2.9 Pg (one billion tonnes) C each year, a quantity that represents the differences between global annual terrestrial production and respiration [5,7].

A major part of this carbon is associated with sediments mobilized through surface runoff. Only a fraction of this carbon is transported directly into the oceans (0.9 Pg C y$^{-1}$) while the majority is mineralized or out-gassed to the atmosphere (1.4 Pg C y$^{-1}$), or buried in lakes and reservoirs (0.6 Pg C y$^{-1}$) [7,8]. Indeed, the ability of natural ecosystems to sequester significant amounts of OC provides a good example of an ecosystem service that can be used in climate mitigation programs on local and regional scales. These mitigation programs may reduce the potential impact of increasing carbon dioxide concentrations in the atmosphere, which are directly and indirectly driving climate change.

Like many ecosystems, river environments around the world are threatened by climate changes [9]. Global changes in rivers include, but are not restricted to, water flow interruptions, temperature increases, changes in nutrient loads, an effluent of new chemicals, occurrence of invasive species and biodiversity losses [9,10]. All of them affect the structure and functioning of the river ecosystem, and thereby their ecosystem services [8,9]. Notwithstanding, increasing concentrations of aquatic carbon is one of the critical changes in river ecosystems [11–13]. Research in this regard mostly focuses on ocean acidification, which is currently rising as a consequence of human activities' emission [14]. However, the effects of elevated atmospheric $CO_2$ levels on freshwater $CO_2$ have not been clearly demonstrated [12]. The degradation of dissolved organic carbon (DOC) has been mentioned as a potential driver of $CO_2$ concentrations in freshwater [15]. The increased DOC concentration can have multiple effects on aquatic macrophyte productivity [16], and hence on the entire food web and ecosystem. Aquatic macrophytes play a crucial role in carbon storage in aquatic systems. However, aquatic macrophytes are affected by increasing carbon concentrations which are related to rising $CO_2$ emissions, which increase organic matter production by aquatic primary producers [17].

Sile River (North Eastern Italy) is the longest (95 km) and the sole fully flat resurgence river in Europe, with a basin area of about 755 km$^2$, located between the basins of the Brenta and Piave rivers (Veneto Region Italy) [18–20]. This river originates from resurgent areas which are located between Padua and Treviso Provinces in North Eastern Italy [21] and flows into the Northern Adriatic Sea [22]. The river is affected by several anthropogenic pressures, such as the channel's modification by different human activities, discharge of agricultural and urban wastewater and the intrusion of alien species, like *Procambarus clarkii* Girard, 1852 and *Silurus glanis* Linnaeus 1758 [23]. The peculiar characteristics of water flows (in normal conditions ranging from 25–30 to 35–45 m$^3$/s) and temperature (12–14 °C) [24] lead to the foundation of an environment with high naturalistic value, but also refuge for all the spontaneous flora of the territory, in a balance between submerged and riparian species and offering food and shelter to animals [25]. The vegetation in the riparian areas has been affected by the composition and inclination of the riverbanks and the environmental conditions. Various aquatic plants occur and change along the river, e.g., *Vallisneria spiralis L.*, *Elodea canadensis Michx.*, *Callitriche stagnalis Scop.* and *Potamogeton natans L.*

Research on river OC stock is very scarce and most studies come from South and North America, with limited studies in Europe [26,27]. Thus, quantifying the OC stocks in these ecosystems may help to define the baseline conditions, which can be further used to address the impact of climate change on OC stocks in aquatic ecosystems. The goals of this work are (i) to quantify the Sile River ecosystem OC stock of sediment and dominant aquatic macrophytes during two sampling periods: summer and winter; (ii) to evaluate the river OC stock in comparison to other ecosystems; (iii) to evaluate the effect of environmental parameters on the ecosystem carbon storage capacity.

## 2. Materials and Methods

### 2.1. Study Area

Submerged aquatic macrophytes (SAMs) and sediments were collected from Sile River in two different sampling campaigns: January (winter) and late July (summer) 2019 in order to estimate the variations in OC stock during spring, the season with the highest

primary productivity [28,29]. Three stations were selected along the Sile River according to their characteristics (Figure 1):

- Storga: a resurgent station with spring characteristics and clear shallow water (ca. 7.0 m transept and 0.5 m depth). Storga is located inside the park "Parco dello Storga" of about 58 ha. The park is characterized by riparian formations of willows (*Salix sp.*), poplars (*Populus sp.*) and alder (*Alnus glutinosa L.*) and by the presence of fountains, streams, and flooded lowlands and hygrophilous woods [30]. The mean water column temperature is 13.7 °C [20];

- Morgano: station before Treviso city centres and located inside the oasis of Santa Cristina. The oasis represents a great marsh of Sile River and a biotope of high naturalistic value with numerous springs. The reserve is part of the Natura 2000 Network as a Site of Community Interest (SCI) and a Special Protection Area (SPA). The station is characterized by ca. 10.0 m transept with maximum 2.5 m water depth. The mean water column temperature is 13.7 °C [20];

- Treviso: is located within the town of Treviso (ca. 25.0 m transepts and ca. 2.5 m depth) with anthropogenic impacts from densely populated neighbourhoods and commercial activities and near a hydroelectric plant. The mean water column temperature is 13.9 °C [20].

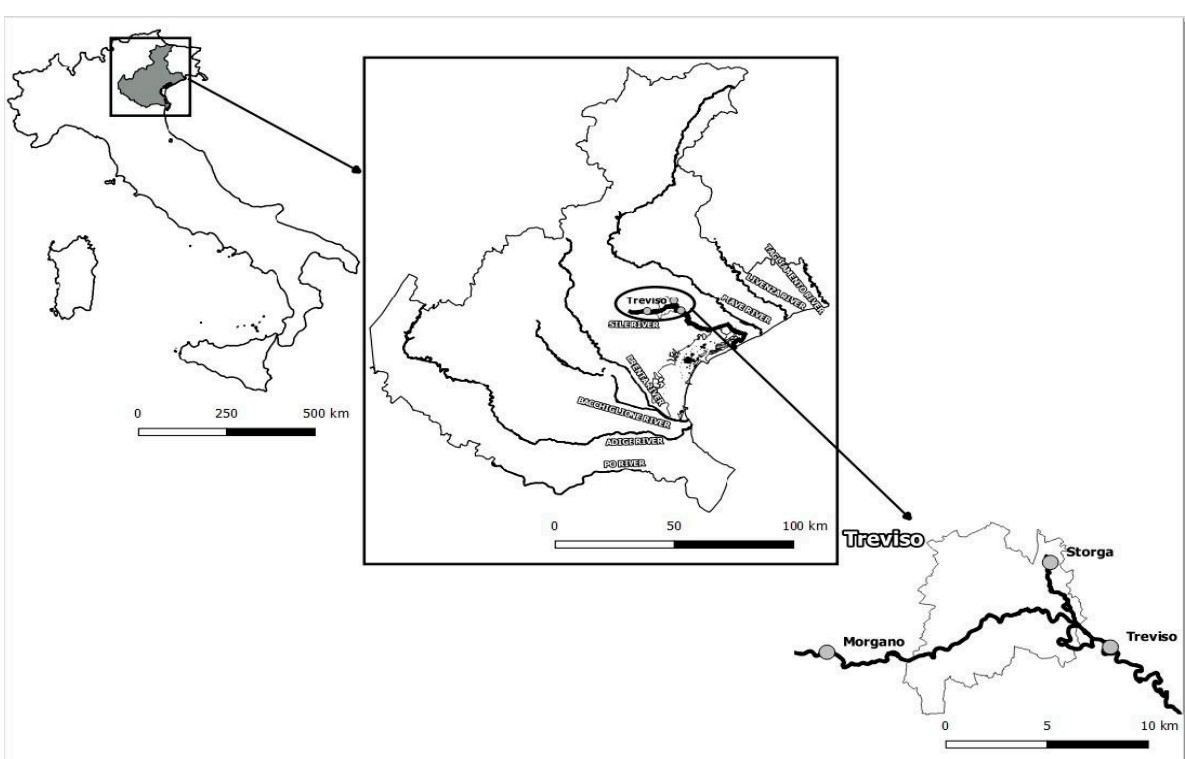

**Figure 1.** Sampling stations along Sile River, North Eastern Italy (Storga, Morgano and Treviso).

### 2.2. SAM Classification and Carbon Quantification

Samples of SAMs were collected in 5 m transept at each station along Sile River in order to calculate the biomass (kg/m$^2$ wet weight) for each species according to Madsen and Wersal [31]. SAMs were collected using a rake within a square of $50 \times 50$ cm$^2$ and kept separately in plastic bags [32]. In laboratory, SAMs were identified according to Pignatti et al. [33]. After separating roots and rhizomes (below-ground part) from leaves (aboveground part), organic carbon (OC) contents were measured using a CHNS elemental analyser (Elementar Vario MICRO Tube, precision <0.1% abs.) after sample parts' pulverizing [34,35]. All analyses were performed in duplicate on different days to obtain a

coefficient of variation <5%. The concentrations of OC are reported in Mg C ha$^{-1}$, after considering the cover areas of each SAM species.

### 2.3. Sediments Analyses

Sediments were sampled using a Plexiglass corer (i.d. 10 cm); each sample was divided into four different subsamples: 0–5, 5–15, 15–25 and 25–35 cm. Three replicates for each subsample were mixed and sorted for OC analyses and the determination of grain-size (% ≤ 63 μm) and density (g/cm$^3$). The sediment pH and Eh were determined using a portable pH-meter (pHenomenal pH 1100 H, VWR, pH accuracy ± 0.005 pH, Eh accuracy ± 0.3 mV).

Organic carbon (OC) was determined indirectly as differences in Total Carbon (TC) and Inorganic Carbon (IC). Total carbon was determined directly by CHNS Elemental Analyser after sediment pulverization with a sediment mill. Inorganic carbon concentration was determined after sample combustion for two hours at 440 °C [36]. All analyses were performed in duplicate in different days to obtain coefficient of variation (CV) ≤5%. The concentrations of sediment OC were reported in Mg C ha$^{-1}$. The percentage of Fines (fraction ≤ 63 μm) was obtained by wet sieving approx. 50 g of dried sediment throughout Endecotts sieves (ENCO Scientific Equipment, Spinea, Italy), after removing shell fragments [37]. Sediment dry density (g/cm$^{-3}$) was determined by gravimetric method weighting the sediment before and after sediment desiccation at 110 °C [38].

### 2.4. Water Environmental Parameters

Temperature (°C), redox potential (Eh) (accuracy ± 0.15%) and pH (accuracy ±0.015 units) were determined using a portable pH meter (pHenomenal, pH 1100 H, VWR,). Dissolved oxygen was determined using an oxygen portable meter (WTW ProfiLine™ Oxi 3310, accuracy ±0.01 mg/L, White Plains, NY, USA). Three surface water samples were collected. Subsamples were filtered through GF/F Whatman glass microfiber filters (porosity 0.7 μm). Filtered waters were analyzed for nutrient (ammonium, nitrite, nitrate, reactive phosphorus, silicate) determination according to the spectrophotometric analysis procedures reported by Strickland and Parson [39]. The quantification at different wavelengths of phosphates, silicates and inorganic nitrogen compounds occurred by comparison with calibration curves made with sea water. All colorimetric analyses were performed in triplicate. *Chlorophyll-a* (Chl-a) concentrations were measured by spectrophotometric analysis [40] using the acetone (90%) extraction protocol. Two additional 500 mL water sub-samples were filtered in situ through GF/F Whatman glass microfiber filters (porosity 0.7 μm), previously dried at 110 °C for two hours, in order to obtain the concentration (mg L$^{-1}$) of the total suspended sediment (TSS), as described previously by Sfriso et al. [41].

Visualization of data was performed using R software (version 3.5.3) and R studio [RStudio, PBC, Austria] (version 1.2.5033) using the packages: ggplot2; ggpubr; RColorBrewer. The study stations map was created using software QGIS (version 2.18.28).

## 3. Results

### 3.1. SAMs Community Composition and Biomass

The in-field sampling revealed that eight species of SAMs were present along the Sile River during the two monitoring periods (summer and winter) in the sampling stations (Table 1). In total, two alien species were identified: *Elodea canadensis* and *Ceratophyllum demersum L.*

**Table 1.** List of submerged aquatic macrophytes (SAMs) and Biomass (kg/m$^2$ wet weight).

| | Summer | | | Winter | | |
|---|---|---|---|---|---|---|
| | Storga | Morgano | Treviso | Storga | Morgano | Treviso |
| **Species** | kg/m$^2$ w.w. | | | | | |
| *Berula erecta* | | 2.53 | 2.27 | | 1.27 | 1.15 |
| *Callitriche brutia* | | 3.81 | 3.60 | | 4.87 | 2.08 |
| *Callitriche stagnalis* | 5.39 | | | 4.60 | | |
| *Ceratophyllum demersum* | 3.95 | 3.85 | 3.19 | 3.91 | 2.75 | 2.58 |
| *Elodea canadiensis* | | 0.54 | 0.49 | | 0.56 | 0.43 |
| *Myosotis scorpioides* | 4.34 | | | 4.17 | | |
| *Myriophyllum spicatum* | | | 1.18 | | | 1.12 |
| *Stuckenia pectinata* | | 5.12 | 2.45 | | 1.41 | 4.91 |
| *Vallisneria spiralis* | | 5.56 | 5.27 | | 1.43 | 1.04 |

The station at Treviso was dominated by all the listed species, except *Myosotis scorpioides L.* and *Callitriche stagnalis Scop.* Instead, the latter were present in the SAM community at Storga. *Myriophyllum spicatum L.* was found only at Treviso, with an annual average biomass of approx. 1.15 kg/m$^2$ whereas *Ceratophyllum demersum* was collected from all stations, with the highest biomass (3.95 kg/m$^2$) recorded at Storga during the summer period. The biomass of this species decreased during winter and showed the lowest biomass (2.58 kg/m$^2$) at Treviso.

The lowest vegetation biomass was represented from Elodea canadensis, with values <0.6 kg/m$^2$ both at Morgano and Treviso. The dominant SAM recorded at Treviso during summer was *Vallisneria spiralis L.* (5.26 kg/m$^2$), while in winter *Stuckenia pectinata (L.)* Börner showed the highest biomass (4.91 kg/m$^2$). Conversely, at Morgano during the summer, the SAM with the highest biomass was *Stuckenia pectinata* (5.11 kg/m$^2$) while in winter the most species abundant was *Callitriche brutia* Petagna (4.86 kg/m$^2$). At Storga *Callitriche stagnalis*, *Myosotis scorpioides* and *Ceratophyllum demersum* represented ca. 95% of the total biomass. Indeed, the biomass of these species did not show significant seasonal variations. For instance, the biomass of *Myosotis scorpioides* and *Ceratophyllum demersum* decreased by 4% and 1%, respectively, from summer to winter. Regarding the biomass variability, the species that had the greatest loss (>70%) during the winter were: *Stuckenia pectinata* at Morgano and *Vallisneria spiralis* at Morgano and Treviso.

*3.2. Water and Sediment Parameter Variations*

The variations in the water column parameters of the three stations are shown in Table S1. The mean concentration of Reactive Phosphorus (RP) in all the stations during the two seasons was of 0.78 ± 0.38 μM, with the highest concentration at Morgano in summer (1.23 μM). On the contrary, the lowest RP values were measured at Treviso (0.31 and 0.30 μM in summer and winter, respectively). Regarding DIN (Dissolved inorganic nitrogen), the highest concentrations were measured at Storga (77.0 μM in summer, 81.2 μM in winter). The concentrations of DIN increased during the cold months at all stations. However, in accordance with the lowest RP values, DIN concentrations at Treviso were also the lowest (mean value: 63.5 μM).

The pH and DO values in the water column highlighted the low variability between the stations and the sampling periods. Indeed, the range of pH was 7.6–7.9 and the range of DO was 10.1–14.5 mg/L, whereas the Chl-a showed the highest concentration at Morgano (mean: 2.40 μg/L) and Storga (mean: 1.84 μg/L) in comparison to Treviso (average: 1.03 μg/L). The clear water recorded at Storga displayed the lowest TSS concentration (4.85 mg/L). This value increased at Morgano (9.65 mg/L) and Treviso (15.7 mg/L).

Regarding the sediment (Table S1), pH ranged between 7.04 and 7.40 in all stations with a mean value of 7.22. The redox potential (Eh) ranged from 82.3 to 228 mV with the lowest values at Treviso. The highest percentage of Fines (Table S2) was measured at Storga (87% for the top 35 cm), while at Treviso the percentage of Fines decreased to 67%. In contrast, Morgano showed the sediment with the lowest percentage of Fines (44%). Moreover, the highest Fine percentage was found at 25–35 cm depth at all stations. Regarding the total nitrogen (TN) concentration (Table S3) in the top 35 cm sediment (4.18–5.87 Mg TN ha$^{-1}$), higher values were observed during summer with little variability between stations.

### 3.3. Sediment Carbon Determination

The distribution of OC in the sediment profile of Sile River sampling sites is shown in Figure 2. The average OC concentration (0–35 cm) was 86.6 Mg C ha$^{-1}$. The highest OC content (0–35 cm) was recorded at Storga during the two sampling periods (104.7 and 104.2 Mg C ha$^{-1}$ in summer and winter, respectively), whereas the lowest content was recorded at Treviso (71.8 Mg C ha$^{-1}$) in winter. Morgano showed an increasing trend from 75.8 to 84.3 Mg C ha$^{-1}$, whereas at Treviso OC decreased from 79.8 to 71.8 Mg C ha$^{-1}$.

The OC content in the sediment 5 cm top layer in all sampling stations ranged between 10.5 and 17.5 Mg C ha$^{-1}$, increasing with the sediment depth. The maximum content was recorded at 25–35 cm depth (21.7–33.7 Mg C ha$^{-1}$).

In the 5 cm top layer at Storga in winter, OC was 17.6 $\pm$ 3.75 Mg C ha$^{-1}$ and increased to 33.7 $\pm$ 3.41 Mg C ha$^{-1}$ in the 35 cm sediment depth, whereas in summer the OC content increased from 12.2 $\pm$ 4.86 Mg C ha$^{-1}$ (0–5 cm depth) to 32.3 $\pm$ 2.89 Mg C ha$^{-1}$ (25–35 depth). Even the OC content at Treviso and Morgano increased with sediment depth (5 to 35 cm) by ca. 200%.

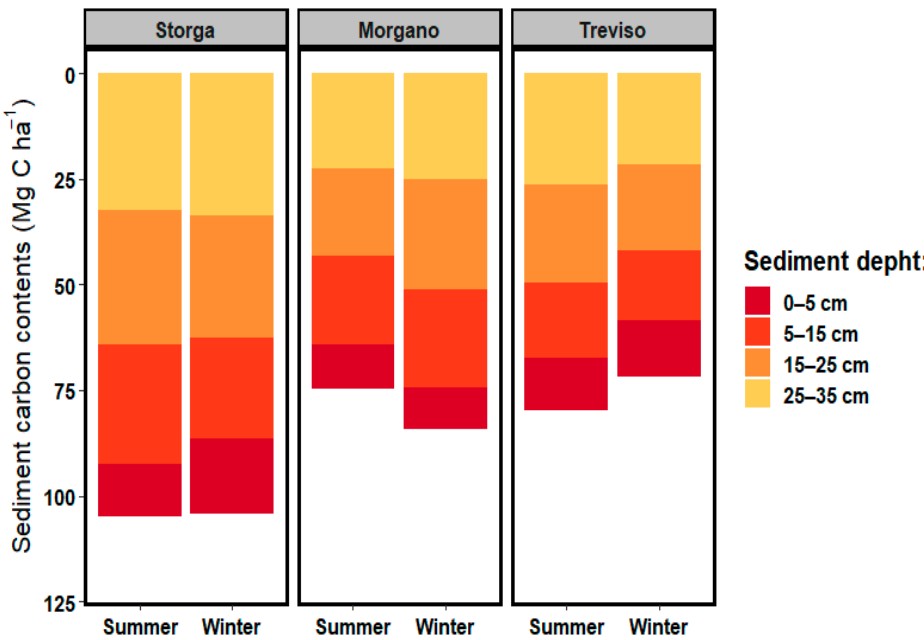

**Figure 2.** Distribution of sediment OC content for Sile River sampling stations.

### 3.4. SAMs Carbon Determination

The total aboveground biomass stored ca. 90% of the SAM carbon stock (Figure 3). The mean OC stored by *C. stagnalis* at Storga was 3.83 Mg C ha$^{-1}$ (aboveground: 3.41 Mg C ha$^{-1}$, belowground: 0.4 Mg C ha$^{-1}$). However, the OC stored in this species decreased to 2.79 Mg C ha$^{-1}$ (aboveground: 2.64 Mg C ha$^{-1}$, belowground part: 0.15 Mg C ha$^{-1}$) at Morgano and to 2.12 Mg C ha$^{-1}$ (aboveground: 1.91 Mg C ha$^{-1}$, belowground:

0.21 Mg C ha$^{-1}$) at Treviso. Seasonally, the OC stored in *C. brutia* increased in the summer period by 11.6%.

Regarding *S. pectinata*, the mean OC storage at Treviso was 2.48 Mg C ha$^{-1}$ (aboveground: 2.28 Mg C ha$^{-1}$, belowground: 0.19 Mg C ha$^{-1}$), while at Morgano OC was 1.82 Mg C ha$^{-1}$ (aboveground: 1.62 Mg C ha$^{-1}$ and belowground 0.20 Mg C ha$^{-1}$). However, the OC stored in this species reduced during the winter period by 64.4%. *Ceratophyllum demersum* was recorded at all stations with a mean OC content of 1.78 ± 0.28 Mg C ha$^{-1}$. *Vallisneria spiralis* and *B. erecta* were only present at Treviso and Morgano and stored 1.09 ± 0.65 and 1.21 ± 0.27 Mg C ha$^{-1}$, respectively. *Myosotis scorpioides* accumulated 24.3% of the total OC (mean value: 1.88 Mg C ha$^{-1}$, aboveground: 1.65 Mg C ha$^{-1}$, belowground: 0.22 Mg C ha$^{-1}$) stored in SAMs at Storga. Some species exhibited low levels of OC storage (<1 Mg C ha$^{-1}$), for instance, *E. canadiensis* averagely stored 0.20 Mg C ha$^{-1}$ at Treviso and Morgano and *M. spicatum* accumulated 0.49 Mg C ha$^{-1}$ at Treviso.

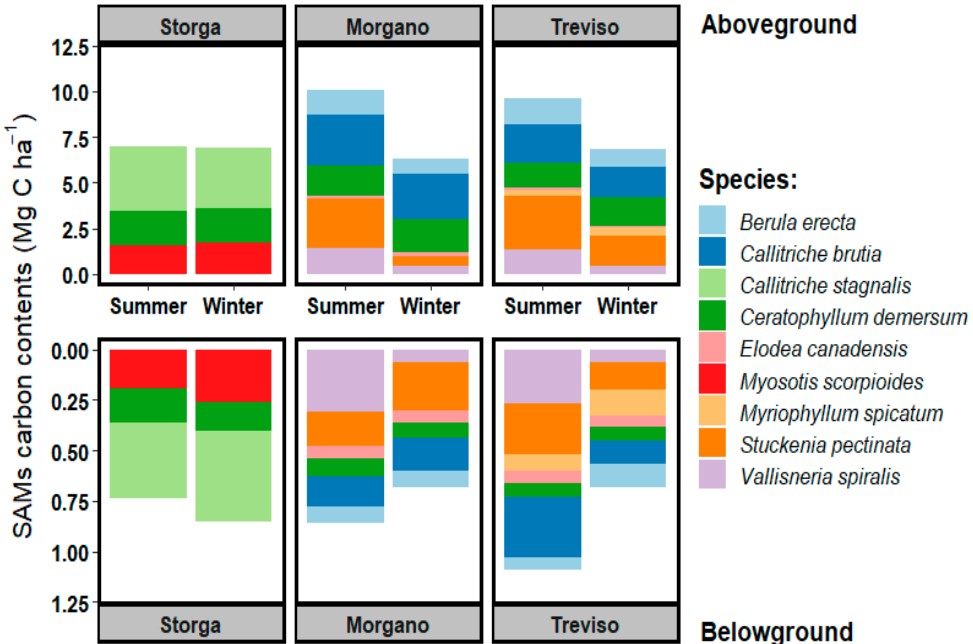

**Figure 3.** Distribution of OC storage in SAMs in the Sile River sampling stations (Aboveground [range: 0–12.5 Mg C ha$^{-1}$] and belowground [range: 0–1.25 Mg C ha$^{-1}$]).

### 3.5. Ecosystem OC Stock

The distribution of the OC stock of the Sile river ecosystem (both sediment and SAMs) is shown in Figure 4. The mean ecosystem OC was 96.2 ± 14.3 Mg C ha$^{-1}$ in summer and slightly lower in winter (94.2 ± 16.4 Mg C ha$^{-1}$). The sediment contributed with 86.6 Mg C ha$^{-1}$, while SAM biomass with 8.66 Mg C ha$^{-1}$ accounting for 9.9% of the total ecosystem OC stock. Storga showed the highest level of OC (112.2 Mg C ha$^{-1}$, i.e., 104.5 and 7.7 Mg C ha$^{-1}$ in sediments and SAMs, respectively), without seasonal differences. In the other two stations (Morgano and Treviso) OC stocks were 88.4 and 85.0 Mg C ha$^{-1}$, respectively, with slight decreases in winter (6.3% and 12.2%, respectively). In these stations the sediments fraction was approx. 90% of the total.

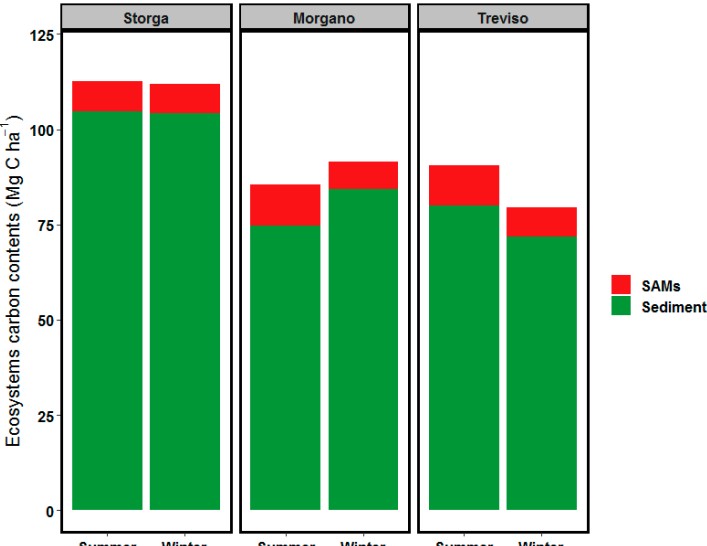

**Figure 4.** Distribution of ecosystem carbon stock for Sile River sampling stations.

## 4. Discussion

### 4.1. SAM Carbon Estimation and Environmental Considerations

This study highlights the capacity of the Sile River ecosystem to store OC in sediments and aquatic plants. The mean ecosystem OC stock ranged from 79.4 to 112.5 Mg C ha$^{-1}$ with no significant seasonal variations among stations. This can be related to the low variability of vegetation biomass among seasons. The SAM accounted for ca. 10% of the ecosystems OC with a mean value of 8.6 ± 1.7 Mg C ha$^{-1}$. Furthermore, the total aboveground biomass accounted for 90% (8.9 ± 1.6 Mg C ha$^{-1}$) of the vegetation OC stock, contrary to the seagrass meadows [42] and mangroves [43], which had the largest proportion of OC stock in belowground parts.

The highest OC amount in the Sile River was stored in the sediment; this could be due to the high sediment capacity for C sequestration [44]. In addition, the highest OC stock was recorded at Storga, which is a resurgent zone of shallow water with spring characteristics influenced by groundwater discharges developed both by springs feeding local sinkholes and waterways sourced inland [30]. As a result, the site is characterized by the finest sediment grain-size and relatively high sediment C:N ratio. The increased storage of carbon in the sediment at Storga could be related to the high affinity of carbon for fine sediments [45]. On the contrary, the sediment OC stock at Morgano and Treviso was lower because sediments were coarser, as previously described by Kauffman et al. [43].

The vegetation composition and structure of river ecosystems is determined by a combination of environmental parameters, including topography, water depth, salinity, nutrient concentration, sediment type, and inter-vegetation competition [46]. Indeed, the seasonal variations in the DO and DIN of water at Morgano and Treviso which were dominated by *S. pectinata* and *V. spiralis* could result in higher OC storage variability among the sampling periods, with a 25% reduction in OC during winter. Moreover, the SAM biomass in these stations increased from winter to summer in agreement with the increasing OC stock of SAMs. Similarly, the size and morphometry of water bodies influenced the SAM distribution, which was affected by the concentrations of DO and pH in water column [47]. There was no clear relationship between OC stock in SAMs and C and N concentrations in sediments. Equally important, several other unmeasured factors were assumed to influence variations in the river carbon stock. For instance, phytoplankton and epiphytes probably had an influence on the water quality and may have been responsible for the SAM biomass changes as recorded from Frodge et al. [47]. These authors showed that growing photosynthesizing plants within the water column contributed to higher mid-water column DO concentrations within patches of *E. canadensis* and *C. demersum*.

The low water flow and lower TSS at Storga may favor the development of SAMs of a distinctive vegetation assemblage, like *Myosotis scorpioides* and *Callitriche stagnalis*. In this context, Chambers et al. [48] showed that the biomass and shoot density of SAMs significantly decreased with the increasing current velocity within the weed beds. Indeed, the lowest SAM diversity and highest biomass were observed at Storga which was characterized by low water flow. It is not surprising that *M. scorpioides*, which was only recorded at Storga, is commonly present in freshwater marshes and slow-moving water. Furthermore, this species grows best in saturated soils with high carbon content and is usually found growing with many obligate plant species such as water parsley (*Oenanthe sarmentosa* C. Presl ex DC, [49]).

The highest OC accumulation at Treviso and Morgano was recorded in *Callitriche brutia* Petagna and *Stuckenia pectinata*. In contrast, *V. spiralis* accumulated lower OC during summer despite the highest biomass obtained for this species in the same period. In fact, the experimental evidence of a study by Coa and Ruan, [50] to investigate how *V. natans* (Hydrocharitaceae) responded to high temperature and $CO_2$, showed that the increase in these parameters significantly enhanced the photosynthetic performance, growth and clonal propagation of *V. natans*.

Moreover, it is important to state that the ecological status of the study stations was in accordance with the Water Framework Directive (2000/60/EC), as previously described by ARPAV, 2019 [20]. The station of Storga was classified as an area with "Good" ecological status, whereas the stations of Treviso and Morgano were assessed as "moderate". Indeed, this highlighted that the ecological status of the aquatic ecosystems might be related with the ecosystem carbon stock capacity, as previously described by Buosi et al. [51] for the sediment Carbon concentration in the seagrass meadows as bioindicator for the ecological status of the Lagoon of Venice [52].

*4.2. Ecosystems Carbon Stock Comparison*

River ecosystems stored a significant proportion of terrestrial carbon stock [2]. The studies that quantify the carbon stored in the River ecosystems are scarce [2,7,45]. The capacity of ecosystems to store carbon is widely variable and related to the primary productivity of ecosystem components [45]. Interestingly, the estimated carbon stock of the Sile river ecosystem was comparable with the other ecosystems' capacity to store carbon. In particular, the seagrass ecosystem had a globally carbon storage capacity ca. 200 Mg C ha$^{-1}$ [42]. This was related to the high primary production of seagrass meadows and their capacity to filter out particles from the water column and store them in sediments [53]. However, the mean carbon stock of seagrass biomass was $7.29 \pm 1.52$ Mg C ha$^{-1}$, which is low when compared with that in forests, which ranges from 30 Mg C ha$^{-1}$, for boreal tundra woodlands, to 300 Mg C ha$^{-1}$ for tropical rainforests [54]. Similar values were obtained in the wetlands in South Eastern Australia [45]. Shallow and deep freshwater marshes showed a mean carbon stock of $200 \pm 20$ and $230 \pm 190$ Mg C ha$^{-1}$, respectively. The permanent open freshwater areas, however, exhibited a mean OC of $110 \pm 12$ Mg C ha$^{-1}$ for the same study [45]. Carbon is accumulated mainly in mangrove and salt marsh ecosystems, in comparison to Sile river ecosystem. Kauffman et al. [43] showed that the mean ecosystem carbon stock in salt marshes east of the mouth of Amazon river (Brazil) was 257 Mg C ha$^{-1}$, whereas the mean mangrove carbon stock was 361–746 Mg C ha$^{-1}$. Similarly, the estimated carbon stock of the mangrove ecosystem within the delta of the Zambezi River (Mozambique) ranged between 373 and 620 Mg C ha$^{-1}$ [55]. In the tropical riverine wetlands of the southern pacific coast of Mexico, Adame et al. [56] estimated that the carbon stock of the mangrove ecosystem was 4-fold higher than the average carbon stock of the Sile river, whereas salt marshes had a mean carbon stock of 336 Mg C ha$^{-1}$.

## 5. Conclusions

This study highlights the contribution of the river ecosystems in the global carbon stock. The quantification of stored OC within sediment and SAMs from the Sile River

implied that the storage capacity for Carbon of the river ecosystem was varied, depending on the differences of vegetation biomass and SAMs biodiversity. Furthermore, the variability of water and sediment environmental parameters were found to affect the ecosystem OC storage capabilities. These findings shed light on the role freshwater rivers play in the global carbon cycle. Indeed, the estimated OC storage capacity from this preliminary research showed that river ecosystems, like Sile River, can significantly contribute to the global assimilation and storage of carbon dioxide. The conservation and protection of river ecosystems have the capacity to reduce the emissions of greenhouse gas and increase the storage of carbon stock while delivering key ecosystem services to river communities. For this reason, the study and protection of river ecosystems and the conservation of SAMs, threatened by anthropogenic pressures and climate change, is fundamental for environmental improvement and ecosystem services. Furthermore, future studies with additional sites and seasonal surveys of the river will enhance our understanding of the effects of global climate changes on the river ecosystem and improve the ecosystem services.

**Supplementary Materials:** The following are available online at https://www.mdpi.com/2073-4441/13/1/80/s1, Table S1: Water and sediment environmental parameters; Table S2: Sediment grain size distribution with depth, Table S3: Sediment total nitrogen (TN) concentrations with depth.

**Author Contributions:** Conceptualization, A.B. and A.S.; methodology and experiments, A.-S.J., A.B. and Y.T.; data analyses and software, A.-S.J., A.B. and Y.T.; writing—original draft preparation, A.B., A.-S.J. and Y.T.; writing—review and editing, A.S., A.B., A.-S.J. and Y.T. All authors have read and agreed to the published version of the manuscript.

**Funding:** This research received department grant from the department of environmental sciences, informatics and statistics, Ca' Foscari University of Venice, Italy.

**Data Availability Statement:** The data presented in this study are available on request from the corresponding author. The data are not publicly available due to privacy.

**Acknowledgments:** The authors would thank the Regional Natural Park (Sile River).

**Conflicts of Interest:** The data of this paper are original, and no part of this manuscript has been published or submitted for publication elsewhere. The authors declare no competing financial interest.

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
