# Peer review of "Ecosystem Organic Carbon Stock Estimations in the Sile River, North Eastern Italy"

_water, doi:10.3390/w13010080_

Round 1

Reviewer 1 Report

Recommendation: MINOR REVISION

Comments to Author:

Ms. Ref. No.: ID: water-1039201

Title: Ecosystem Organic Carbon Stock Estimations in the Sile River, North Eastern Italy

Overview and general recommendation:

I recommend that a minor revision is warranted. I explain my concerns in more detail below. I ask that the authors specifically address each of my comments in their response.

  1. This study aims as objectives: to quantify the Sile River ecosystem OC stock of sediment and dominant aquatic macrophytes during two sampling periods: summer and winter; to evaluate the river OC stock in comparison to other ecosystems; to evaluate the effect of environmental parameters on the ecosystem carbon storage capacity. The current study is on a topic of relevance and general interest to the readers of the journal with relative impact on scientific research, being considered rather a particular case study (from my point of view is more a particular case with impact only for the authorities from Italy - only two monitoring campaign/year is not enough for a better understanding).
  2. In general, a clear thread should be woven through the text linking the introduction, methods, results, and conclusions. After carefully checking the article, I would suggest the following improvements to the article.
  • First of all, some phrases/sentences/ are not clear, some words are not properly used. Please do not use the first person in the manuscript (i.e., we quantify (line 11), we aimed (line 81), …etc..); instead, use the passive voice! Please check in the whole article!
  • The abstract must be rewritten/reformulated (lines 9-14)! The novelty of the study is not clearly explained. The aim of this work at the end of the introduction is unclear and must be correlated with the novelty of this study. Please provide a clear objective/hypothesis in the Abstract and Introduction Section, as well!
  • In Conclusion, please clarify the rationale and reasons for the used methods and the interest to the readers! Explain how other readers can use this work in the future!
  1. Table 1. – please correct Kg/m2 wet weight as [kg/m2 w.w.]. w.w. must be explained in the main text. According to IUPAC kilogram (unit of measurement, UM) is correct kg, not Kg! Please correct in the whole MS!
  2. Lines 179 – 184: mg/l must be replaced with mg/L according to the rules of IUPAC!
  3. Raise the quality of English in the manuscript!

Author Response

Answers to Reviewers Comments

Response to Anonymous Referee #1:

The authors would like to thank the reviewer for precise and thoughtful comments and constructive criticism which has led to a better manuscript. Below we respond to referee comments individually.

The responses to the individual comments of the reviewers are detailed below.
(Note: Reviewer comments are in Bold; authors' responses are in italics red text).

  1. This study aims as objectives: to quantify the Sile River ecosystem OC stock of sediment and dominant aquatic macrophytes during two sampling periods: summer and winter; to evaluate the river OC stock in comparison to other ecosystems; to evaluate the effect of environmental parameters on the ecosystem carbon storage capacity. The current study is on a topic of relevance and general interest to the readers of the journal with relative impact on scientific research, being considered rather a particular case study (from my point of view is more a particular case with impact only for the authorities from Italy- only two monitoring campaign/year is not enough for a better understanding).

Answer: The authors would thank the reviewer for this clarification and his effort to improve the manuscript. Indeed, the scope of this preliminary study was to quantify the Sile River ecosystem carbon stock within the two campaign because we considered the annual growing trend of the common submerged aquatic plants at Sile River in order to highlight the highest primary production period of the year. On the other hand, similar to the Lagoon of Venice (Italy), Sfriso et al., 2020 indicated that the environmental parameters for sediment did not considerably change along the year.    

  1. In general, a clear thread should be woven through the text linking the introduction, methods, results, and conclusions. After carefully checking the article, I would suggest the following improvements to the article.
  • First of all, some phrases/sentences/ are not clear, some words are not properly used. Please do not use the first person in the manuscript (i.e., we quantify (line 11), we aimed (line 81), …etc..); instead, use the passive voice! Please check in the whole article!

    Answer: all the phrases of the first person were removed and we add the correct phrases.  

  • The abstract must be rewritten/reformulated (lines 9-14)! The novelty of the study is not clearly explained. The aim of this work at the end of the introduction is unclear and must be correlated with the novelty of this study. Please provide a clear objective/hypothesis in the Abstract and Introduction Section, as well!

Answer: we completely agree with the reviewer about the missing part of the abstract, however, we rephrased and provided clear objective of our work.  

[River ecosystems are one of the dynamic components of the terrestrial carbon cycle that provide crucial function in ecosystem processes and high value to ecosystem services. Large amount of carbon is transported from terrestrial to the ocean through river flows. In order to evaluate the  contribution of Sile River ecosystem on the global carbon stock,  the river ecosystem Organic Carbon (OC) stock was quantified for sediments and dominant submerged aquatic macrophytes (SAMs) during the two sampling periods at three different stations along Sile River (North Eastern Italy)].

  • In Conclusion, please clarify the rationale and reasons for the used methods and the interest to the readers! Explain how other readers can use this work in the future!

Answer: the conclusion part was rephrased and improved.

[This study highlights the contribution of the river ecosystems in the global carbon stock. The quantification of stored OC within sediment and SAMs from the Sile River implied that the storage capacity for Carbon of the river ecosystem was varied, depending on the differences of vegetation biomass and SAMs biodiversity. Furthermore, the variability of water and sediment environmental parameters were found to affect the ecosystem OC storage capabilities. These findings shed light on the role freshwater rivers play in the global carbon cycle. Indeed, the estimated OC storage capacity from this preliminary research showed that river ecosystems, like Sile River, can significantly contribute to the global assimilation and storage of carbon dioxide. The conservation and protection of river ecosystems have the capacity to reduce the emissions of greenhouse gas and increase the storage of carbon stock while delivering key ecosystem services to river communities. For this reason, the study and protection of river ecosystems and the conservation of SAMs, threatened by anthropogenic pressures and climate changes, is fundamental for environmental improvement and ecosystem services. Furthermore, future studies with additional sites and seasonal surveys of the river will enhance our understanding of the effects of global climate changes on the river ecosystem and improve the ecosystem services].

  1. Table 1. – please correct Kg/m2 wet weight as [kg/m2 w.w.]. w.w. must be explained in the main text. According to IUPAC kilogram (unit of measurement, UM) is correct kg, not Kg! Please correct in the whole MS!

Answer: all the units and acronyms were corrected along all the manuscript.

  1. Lines 179 – 184: mg/l must be replaced with mg/L according to the rules of IUPAC!

Answer: the units were corrected accordingly.

  1. Raise the quality of English in the manuscript!

Answer: The authors would like to thank deeply the reviewer for his time and effort. The English structure and grammar of the manuscript has been thoroughly reviewed through a specialized English editing colleague for proof reading. Repetitive words and concepts have been removed and some paragraphs have been rephrased. We hope that the current version of the manuscript has been improved in readability and focuses more on the research goal.

Reviewer 2 Report

The manuscripts provides useful estimates of organic carbon stock in the Sile River, North Eastern Italy. They quantify the river ecosystem organic carbon stock for sediments and dominant submerged aquatic macrophytes during two sampling periods at three stations along Sile River (North Eastern Italy). They provides useful data, and they demonstrates the importance of freshwater river  ecosystems in the global carbon cycle. The manuscript is clear, but a brush up of the English usage is necessary.

Author Response

Answers to Reviewers Comments

Response to Anonymous Referee #2:

The authors would like to thank the reviewer for precise and thoughtful comments and constructive criticism which has led to a better manuscript. Below we respond to referee comments individually.

The responses to the individual comments of the reviewers are detailed below.
(Note: Reviewer comments are in Bold; authors' responses are in italics red text).

The manuscripts provides useful estimates of organic carbon stock in the Sile River, North Eastern Italy. They quantify the river ecosystem organic carbon stock for sediments and dominant submerged aquatic macrophytes during two sampling periods at three stations along Sile River (North Eastern Italy). They provide useful data, and they demonstrates the importance of freshwater river  ecosystems in the global carbon cycle. The manuscript is clear, but a brush up of the English usage is necessary.

The authors would appreciate the reviewer for his positive feedback about our manuscript. The English structure and grammar of the manuscript has been thoroughly reviewed through a specialized English editing colleague for proof reading. Repetitive words and concepts have been removed and some paragraphs have been rephrased. We hope that the current version of the manuscript has been improved in readability and focuses more on the research goal.

Reviewer 3 Report

The authors present a novel study of quantifying carbon pools in an Italian river. The study is well designed, the English language is well written and I think the study will be of interest to the readers of Water. I only have few comments to the manuscript, they are listed below. Chief among my concerns is the lack of references and in particularly the statement that carbon in rivers is poorly investigated.

Line 29: You should specify that its organic carbon only you talk about here. DIC have different pathways into the system.

Line 30-32: I don’t know if I trust that rivers are so uninvestigated as the authors claim. The reference is more than a decade old and a lot of research has gone into rivers since.

Line 51-54: You need references to this mini review of global change impacts in rivers.

Line 78: I am not impressed with this lit review. A google scholar search for “organic Carbon in rivers“ between 2010 and 2020, gave approx. 126.000 hits.

A search for “organic Carbon pools in rivers“ between 2010 and 2020, gave approx. 16.600 hits. From those several studies may also be of interest.

In both cases you get about an order of magnitude more results without limiting yourself to the past decade. Additional interesting searches could be organic carbon stock, organic carbon budgets, some knowledge might also be had from just searching for carbon, without limiting the search to organic carbon, as some studies look at all carbon pools, organic and inorganic. For instance:
https://www.sciencedirect.com/science/article/abs/pii/S1470160X17302352

In short, I don’t think its fair to say that DOC/POC studies in rivers outside of the US is that scarce. And certainly if you want to make this somewhat controversial point, I would back it up with substantial references. I don’t understand why this claim is supported by two studies of terrestrial (forest) systems in Germany and Italy, and not even by any of American studies mentioned in by the authors?

Figure 1, lovely way of presenting the study.

Line 132: probably don’t have to present the same pH meter again, was redox potential also measured in the free water?

Line 134: Add the accuracy for dissolved oxygen as well.

Line 145-146: It might be useful for the reader to know what packages was used.

Table1: I cant see from the methods how wet weight was measured and calculated.

Line 175: Might want to explain what DIN stands for at the first usage of the acronym, like you do with phosphorus three lines above.

Line 210: I had to read this figure several times before I realized that the scale on the Y axis changed from the top panels to the bottom panels. It might be stronger visually if the panels had the same scale. Alternatively, a point could be made in the legend to note the scales on the two sets of panels.

Line 247-248: I think you may have to ease the reader into why you are comparing the northern Italian river system to Amazonian mangroves. Is there any reason we would expect the Italian river to behave similar or differently from the mangroves or what is the reason you highlight this fact?

Line 273: what does “low hydrodynamics” mean? Low water level? Low variability in water level? Something else? Please specify.

Line 288: I agree its important in general, but perhaps not relevant to the present study?

Line 294-297: References missing.

Author Response

Answers to Reviewers Comments

Response to Anonymous Referee #3:

The authors would like to thank the reviewer for precise and thoughtful comments and constructive criticism which has led to a better manuscript. Below we respond to referee comments individually.

The responses to the individual comments of the reviewers are detailed below.
(Note: Reviewer comments are in Bold; authors' responses are in italics red text).

The authors present a novel study of quantifying carbon pools in an Italian river. The study is well designed, the English language is well written, and I think the study will be of interest to the readers of Water. I only have few comments to the manuscript, they are listed below. Chief among my concerns is the lack of references and in particularly the statement that carbon in rivers is poorly investigated.

Line 29: You should specify that its organic carbon only you talk about here. DIC have different pathways into the system

Answer: the authors would thank the reviewer for this clarification. We highlighted that the source of carbon that deliver the river ecosystem include both organic and organic sources. In fact, organic carbon enters the ecosystem by photosynthetic activities. The text was corrected accordingly.   

[Significant amounts of organic and inorganic carbon typically deliver rivers from the surrounding landscape or originated by the photosynthesis of algae and plants in the water [1]. Despite their dynamic role in the terrestrial carbon stock cycle, river systems and the potential mechanistic controls of OC storage are among the least investigated [4], in comparison to other ecosystems].

Line 30-32: I don’t know if I trust that rivers are so uninvestigated as the authors claim. The reference is more than a decade old and a lot of research has gone into rivers since.

Answer: the authors would thank the reviewer  for this point. Indeed, unless the presence of many works about carbon recently, the articles that studied the River carbon stock are rare. Moreover, the existing articles topic meanly concentrated on the river water carbon (DOC, TOC, POC) rather than sediment and aquatic plants.

Line 51-54: You need references to this mini review of global change impacts in rivers.

Answer: the missing references related to the global change impact were added to the text.

Line 78: I am not impressed with this lit review. A google scholar search for “organic Carbon in rivers“ between 2010 and 2020, gave approx. 126.000 hits.

A search for “organic Carbon pools in rivers“ between 2010 and 2020, gave approx. 16.600 hits. From those several studies may also be of interest.

In both cases you get about an order of magnitude more results without limiting yourself to the past decade. Additional interesting searches could be organic carbon stock, organic carbon budgets, some knowledge might also be had from just searching for carbon, without limiting the search to organic carbon, as some studies look at all carbon pools, organic and inorganic. For instance:
https://www.sciencedirect.com/science/article/abs/pii/S1470160X17302352

In short, I don’t think its fair to say that DOC/POC studies in rivers outside of the US is that scarce. And certainly if you want to make this somewhat controversial point, I would back it up with substantial references. I don’t understand why this claim is supported by two studies of terrestrial (forest) systems in Germany and Italy, and not even by any of American studies mentioned in by the authors?

Answer: We thank the reviewer for his effort to improve our manuscript. In this study we try to compare our obtained results (Carbon stock of sediment and submerged aquatic plants) from the study area with other aquatic ecosystems. Indeed, unless the presence of thousands of articles of the topic Carbon [Carbon flux, DOC, DIC], the articles about the topic (Carbon stock of river ecosystems) are scare. Many of the recent published articles discussed the carbon stock in aquatic ecosystems include mangrove, Estuaries, seagrass meadows. Whereas, the published work related to the aquatic plants and sediment in the River are scare, especially in the north of Italy.  

Figure 1, lovely way of presenting the study.

Answer: deep thanks for your opinion.

Line 132: probably don’t have to present the same pH meter again, was redox potential also measured in the free water?

Answer: indeed, the pH and redox potential (Eh) for the water and sediment were measured in the same portable instrument.  

Line 134: Add the accuracy for dissolved oxygen as well.

Answer: the accuracy of the instrument was added into the related text.

Line 145-146: It might be useful for the reader to know what packages was used.

Answer: the name of used packages used to create the figures were added into the text. 

Table1: I cant see from the methods how wet weight was measured and calculated.

Answer: we already highlighted that the method of measurement is reported in mentioned reference  (Madsen, J. D.; Wersal, R. M. A review of aquatic plant monitoring and assessment methods).

Line 175: Might want to explain what DIN stands for at the first usage of the acronym, like you do with phosphorus three lines above.

Answer: the acronym was corrected accordingly.

Line 210: I had to read this figure several times before I realized that the scale on the Y axis changed from the top panels to the bottom panels. It might be stronger visually if the panels had the same scale. Alternatively, a point could be made in the legend to note the scales on the two sets of panels.

Answer: we thank the reviewer for this important not. we specify in the legend of the figure that aboveground and underground parts of SAMs had different scale. [(Aboveground [range: 0-12.5 Mg C ha-1] and belowground [range:0-1.25 Mg C ha-1])].

Line 247-248: I think you may have to ease the reader into why you are comparing the northern Italian river system to Amazonian mangroves. Is there any reason we would expect the Italian river to behave similar or differently from the mangroves or what is the reason you highlight this fact?

Answer: we decided to make a comparison with the mangrove ecosystem because they are both aquatic ecosystems. In addition, we add also new citation about the global seagrass ecosystem to support our results. In fact, its correct that they are in different areas, but in this point of discussion we want to highlight that there are a differences between the carbon stock accumulated in plant compartments (above and below-ground).

Line 273: what does “low hydrodynamics” mean? Low water level? Low variability in water level? Something else? Please specify.

Answer: we changed the term to be more clear for the readers. “the low water flow”

Line 288: I agree its important in general, but perhaps not relevant to the present study?

Answer: We urged to highlight how the Ecological Status (and its evaluation and conservation) is also important to the ecological service of the "Carbon stock”). The below sentence was added:

[Indeed, this highlighted that the ecological status of the aquatic ecosystems might be related with the ecosystem carbon stock capacity as previously described by Buosi et al., [2020] for the sediment Carbon concentration in the seagrass meadows as bioindicator for the ecological status of the Lagoon of Venice [Sfriso et al., 2014]].

Line 294-297: References missing.

Answer: the missing reference was added
